# RELATE: Physically Plausible Multi-Object Scene Synthesis Using Structured Latent Spaces

**Sébastien Ehrhardt** [1]*   **Oliver Groth**[1]*   **Áron Monszpart**[2,3]   **Martin Engelcke**[1]
**Ingmar Posner**[1]   **Niloy J. Mitra**[2,4]   **Andrea Vedaldi**[1]
[1]Department of Engineering Science, University of Oxford
[2]Department of Computer Science, University College London
[3]Niantic, [4] Adobe Research
{hyenal,ogroth}@robots.ox.ac.uk

## Abstract

We present RELATE, a model that learns to generate physically plausible scenes and videos of multiple interacting objects. Similar to other generative approaches, RELATE is trained end-to-end on raw, unlabeled data. RELATE combines an object-centric GAN formulation with a model that explicitly accounts for correlations between individual objects. This allows the model to generate realistic scenes and videos from a physically-interpretable parameterization. Furthermore, we show that modeling the object correlation is *necessary* to learn to disentangle object positions and identity. We find that RELATE is also amenable to physically realistic scene editing and that it significantly outperforms prior art in object-centric *scene* generation in both synthetic (CLEVR, ShapeStacks) and real-world data (cars). In addition, in contrast to *state-of-the-art* methods in object-centric generative modeling, RELATE also extends naturally to dynamic scenes and generates *videos* of high visual fidelity. Source code, datasets and more results are available at http://geometry.cs.ucl.ac.uk/projects/2020/relate/.

## 1   Introduction

We consider the problem of learning to generate plausible images of scenes starting from parameters that are physically interpretable. Furthermore, we wish to learn such a capability from raw images alone, without any manual or external supervision. Image generation is often approached via Generative Adversarial Networks (GAN) [10]. These models learn to map noise vectors, used as a source of randomness, to image samples. While the resulting images are realistic, the random vectors that parameterize them are not interpretable. To address this issue, authors have recently proposed to *structure* the latent space of deep generative models, giving it a partial physical interpretability [28, 29, 36]. For example, HoloGAN [28] samples volumes and cameras to generate 2D images of 3D objects, and BlockGAN [29] creates scenes by composing multiple objects. The resulting GANs have been shown to learn concepts such as viewpoint and object disentangling from raw images.

BlockGAN is of particular interest because, via its relatively strong architectural biases, it provides *interpretable* parameters for the scene, incorporating concepts such as position and orientation. However, BlockGAN comes with a significant limitation in that it assumes that objects are mutually *independent*. This approximation is acceptable only when objects interact weakly, but it is badly violated for medium to densely packed scenes, or for scenes such as stacking wooden blocks or cars following a path, where the (object) correlation is strong.

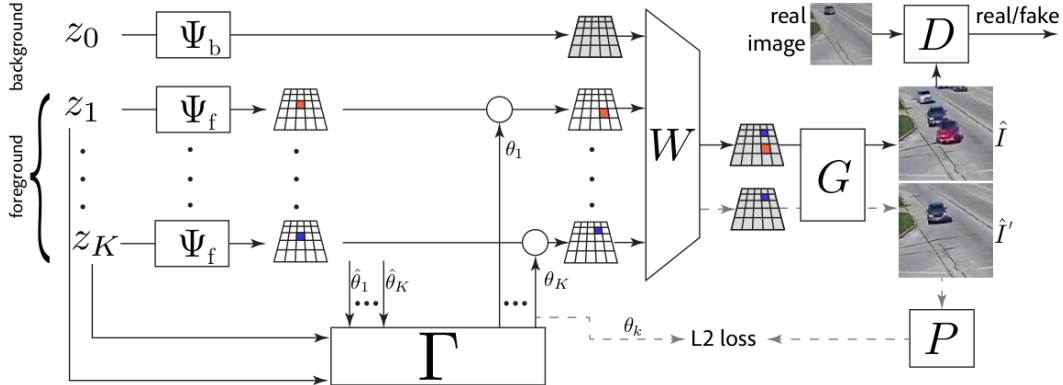

**Figure 1: Image generation using RELATE.** Individual scene components, such as background and foreground objects are represented by appearance $z_0$ and pairs of appearance and pose vectors $(z_i, \theta_i), i \in \{1, \dots, K\}$, respectively. The key spatial relationship module $\Gamma$ adjusts the initial independent pose samples $\hat{\theta}_i$ to be physically plausible (*e.g.*, non-intersecting) to produce $\theta_i$. The structured scene tensor $W$ is finally transformed by the the generator network $G$ to produce an image $\hat{I}$. RELATE is trained end-to-end in a GAN setup ($D$ denotes the discriminator) on real unlabelled images.

Recent work in object-centric generative modeling has attempted to specifically address this by capturing correlations in latent space (*e.g.*, [7, 36]). However as object state information remains significantly entangled in these models they have, to date, been unable to operate on real-world data.

In this paper, we introduce RELATE, a model which explicitly leverages the strong architectural biases of BlockGAN to effectively model correlations between latent object state variables. This leads to a powerful model class, which is able to capture complex physical interactions, while still being able to learn from raw visual inputs alone. Empirically, we show that only when we model such interactions our GAN model correctly disentangles different objects when they exhibit even a moderate amount of correlation (figures 2 and 3). Without this component, the model may still generate high fidelity images, but it generally fails to establish a physically-plausible association between the parameters and the generated images. Our results also demonstrate that GANs are surprisingly sensitive to the correlation of objects in natural scenes, and can thus be used to directly learn these *without* resorting to techniques such as variational auto-encoding (VAE [19]).

We demonstrate the efficacy of RELATE in several scenarios, including balls rolling in bowls of variable shape [6], cluttered tabletops (CLEVR [16]), block stacking (ShapeStacks [12]), and videos of traffic at busy intersection. By ablating the interaction module, we show that modeling the spatial correlation between the objects is key. Furthermore, we compare RELATE to several recent GAN- and VAE-based baselines, including BlockGAN [29], GENESIS [7] and OCF [1], in terms of *Fréchet Inception Distance (FID)* [13], and outperform even the best state-of-the-art model by up to 29 points.

Qualitatively, we show that modeling spatial relationships strongly affects scene decomposition and the enforcement of spatial constraints in the generated images. We also show that the physically interpretable latent space learned by RELATE can be used to edit scenes as well as to generate scenes outside the distribution of the training data (*e.g.*, containing more or fewer objects). Finally, we show that the parameterization can be used to generate long plausible video sequences (as measured according to FVD score [34]) by simulating their dynamics while preserving their spatial consistency.

## 2  Related Work

**Interpretable Object-Centric Visual Models.**    Inspired by the *analysis-by-synthesis* approach for visual perception discussed in cognitive science [44], recent work [3, 7, 11, 36] propose structured latent space models to explain and synthesize images as sets of constituent components which are individually represented using VAEs [19] or GANs [10]. Other approaches favor explicit symbolic representations over distributed ones when parsing an image [33, 40] or propose probabilistic programming languages to formalize image generation [23]. In both cases, object-centric modeling allows decomposition of images into components and also enables targeted image modification via

interpolation in symbol or latent space, *e.g.*, altering position or color of an object parsed by the model. Interpretable and controllable factors of image generation are desirable properties for neural rendering models and have been investigated in recent image generation models, *e.g.*, [1, 25, 28, 29]. However, despite the modeling effort put into the object representations, inter-object interactions are typically only modeled in a less explicit way, *e.g.*, via image layers [36, 40], depth ordering variables [1] or an autoregressive *scene prior* [7]. Our work adds to this line of work by proposing a spatial correlation network which facilitates disentanglement of learned object representations and can be trained from raw observations.

**Neural Physics Approximation.** Harnessing the power of deep learning to approximate physical processes is an emerging trend in the machine learning community. Especially the approximation of rigid body dynamics with neural networks already boasts a large body of literature, *e.g.*, [2, 4, 8, 22, 35, 39]. Such learned approximations of object interactions have been successfully employed in object manipulation [15] and tracking [9, 21]. However, most entries in this line of work are only applied to visual toy domains or rely on segmentation masks or bounding boxes to initialize their object representations before the networks approximate the object dynamics. While we leverage ideas from neural dynamics modeling, we go beyond the established scope of visual toy domains such as colored point masses or moving MNIST digits [24] and learn directly from rich visual data such as simulated object stacks and real traffic videos.

**Extraction and Generation of Video Dynamics.** Videos are a natural choice of data source to learn about the physics of rigid bodies. Physical information extracted from videos can either be explicit such as estimates of velocity or friction [42] or implicitly represented in the latent space, *e.g.*, sub-spaces corresponding to pose variation [5]. More recently, object-centric approaches have also been leveraged to acquire better video representations for future frame prediction [43] or model-based reinforcement learning [37]. Several studies have also attempted to learn entire video distributions as spatio-temporal tensors in GAN frameworks [17, 32, 38, 41] yielding impressive first results for full video generation in artificial and real domains. In contrast to prior art, our model departs from a monolithic spatio-temporal tensor representation over an entire video. Instead we cast the video learning and generation process as temporal extension of the object-centric representation of a single frame, lowering the computational burden while still faithfully representing long-range dynamics.

# 3 Method

RELATE (figure 1) consists of two main components: An interaction module, which computes physically plausible inter-object and object-background relationships, and a scene composition and rendering module, which features an interpretable parameter space factored into appearance and position vectors. The details are given next.

## 3.1 Physically-interpretable scene composition and rendering

RELATE considers scenes containing up to $K$ distinct objects. The model starts by sampling *appearance parameters* $z_1, \ldots, z_K \sim \mathcal{U}([-1,1]^{N_f})$ for each individual foreground object as well as a parameter $z_0 \sim \mathcal{U}([-1,1]^{N_b})$ for the background. These parameters are small noise vectors, similar to the ones typically used in generative networks. Different from the object poses below, they are sampled independently, thus assuming that the appearance of different objects is independent.

For rendering an image, the appearance parameter $z_k$ is first mapped to a tensor $\Psi_k \in \mathbb{R}^{H \times H \times C}$. This is done via two separate learned decoder networks, one for the background $\Psi_0 = \Psi_b(z_0)$ and one for the foreground objects $\Psi_k = \Psi_f(z_k)$. Here $H$ is the horizontal and vertical spatial resolution of the representation (see table A2 in supplementary) and $C$ is the number of feature channels (see tables A3 and A4). Since we assume that individual objects are much smaller than the overall scene, we restrict $\Psi_k, k \geq 1$ to be non-zero only in a fixed smaller $H' < H$ window in the center of the tensor.

Each foreground object also has a corresponding *pose parameter* $\theta_k$, which is geometrically interpretable. For simplicity, we assume $\theta_k \in \mathbb{R}^2$ to be a 2D translation, acting on the tensor $\Psi_k$ via bilinear resampling:

$$\hat{\Psi}_k = \theta_k \cdot \Psi_k \quad \text{such that} \quad [\hat{\Psi}_k]_u = [\Psi_k]_{u+\theta_k}$$

where $u \in \mathbb{R}^2$ is a spatial index and $[\cdot]_u$ means accessing the column of the tensor in bracket at spatial location $u$ (using padding and bilinear interpolation if $u$ does not have integer coordinates). However, $\theta_k$ can easily be extended to represent full 3D transformations as previously shown in BlockGAN [29].

Foreground and background objects are composed into an overall scene tensor $W \in \mathbb{R}^{H \times H \times C}$ via element-wise max- (or sum-) pooling as $W_u = \max_{k=0,\ldots,K} [\hat{\Psi}_k]_u$. In this manner, the scene tensor is a function $W(\Theta, Z)$ of the pose parameters $\Theta := (\theta_1, \ldots, \theta_K)$ and the appearance parameters $Z := (z_0, z_1, \ldots, z_K)$. Finally, a decoder network $\hat{I} = G(W)$ renders the composed scene as an image (see table A5).

**Discussion.** This model is 'physically interpretable' in the sense that it captures (1) the identities of $K$ distinct objects and (2) their pose parameters as translation vectors. This should be contrasted to traditional GAN models, where the code space is given as an uninterpretable, monolithic noise vector $z$. Despite the structure given to the code space, there is no guarantee that the model will actually learn to map it to the corresponding structure in the example images. However, we found empirically that this is the case as long as the correlations between the different objects are also captured.

### 3.2 Modeling correlations in scene composition

RELATE departs significantly from prior art such as BlockGAN as it does not assume the parameters $\theta_i$ of the different objects to be independent. In order to model correlation, we propose a two-step procedure, based on a residual sampler. First, we sample a vector of $K$ i.i.d. poses $\hat{\Theta} \sim \mathcal{U}([-H''/2, H''/2]^{2K})$ where $H'' < H$ is smaller than the spatial size $H$ of the tensor encoding. Then, we pass this vector to a 'correction' network $\Gamma$ that remaps the initial configuration to one that accounts for the correlation between object locations and appearances, as well as between objects and the background (coded by the appearance component $z_0$ in $z$): $\Theta := \Gamma(\hat{\Theta}, Z)$. In practice, we expect object interactions, as any physical law, to be *symmetric* with respect to the order of the objects. We obtain this effect by implementing $\Gamma$ as running $K$ copies of the *same* corrective function in parallel:

$$\theta_k = \hat{\theta}_k + \zeta(\hat{\theta}_k, z_k, |z_0, \{z_i, \hat{\theta}_i\}_{i \geq 1, i \neq k}). \tag{1}$$

The function $\zeta$ is implemented in a manner similar to the Neural Physics Engine (NPE) [4]:

$$\zeta(\hat{\theta}_k, z_k, |z_0, \{z_i, \hat{\theta}_i\}_{i \geq 1, i \neq k}) = f(\hat{\theta}_k, z_k, z_0, h_k^s), \quad h_k^s = \sum_{q \neq k} g(\hat{\theta}_k, z_k, \hat{\theta}_q, z_q), \tag{2}$$

where $f$ and $g$ are Multi Layer Perceptrons (MLPs) (tables A6, A7) operating on stacked vector inputs and $h^s$ is an embedding capturing the interactions between the $K$ objects. Besides symmetry, an advantage of this scheme is that it can take an arbitrary number of objects $K$ due to the sum-pooling operator used to capture the interactions. In this manner, the sampler $\Gamma$ is automatically defined for any value of $K$. For each scene, K is sampled uniformly from a fixed interval $[K_{\min}, K_{\max}]$. Furthermore, sampling independent quantities followed by a correction has the benefit of injecting some variance on the objects positions at the early stage of training, which helps to avoid converging to trivial/bad solutions.

**Ordered scenes.** An advantage of RELATE is that it can be easily modified to take advantage of additional structure in the scene. For scenes where objects have natural order, such as stacks of blocks, we experiment with conditioning pose $\theta_i$ on the preceding pose $\theta_{i-1}$, using a Markovian process. This is done by first sampling $\hat{\theta}_1 \sim \mathcal{U}([-H''/2, H''/2])$, and then applying a correction to account for the background $z_0$ as before, finally sampling the other objects in sequence:

$$\theta_1 = \hat{\theta}_1 + f_0(\hat{\theta}_1, z_1, z_0), \quad \forall k > 1 : \quad \theta_k = \theta_{k-1} + f_1(\theta_{k-1}, z_{k-1}, z_0), \tag{3}$$

where $f_0, f_1$ are implemented as MLPs as before (tables A9, A10). Note that this can be interpreted as a special case of the model above in the sense that we can write $\Theta := \Gamma(\hat{\Theta}, Z)$, provided that $\hat{\theta}_k = 0$ for $k \geq 2$.

**Modeling dynamics.** RELATE can also be immediately extended to make dynamic predictions. For this, we sample the initial positions $\theta_k(0)$ as before and then update them incrementally as $\theta_k(t + 1) = \theta_k(t) + v_k(t + 1)$, where $v_k(t)$ is the object velocity. In order to obtain the latter, we let $V_k(t) = [v_k(t - i)]_{i=2,1,0}$ denote the last three velocities of the $k$-th object. The initial

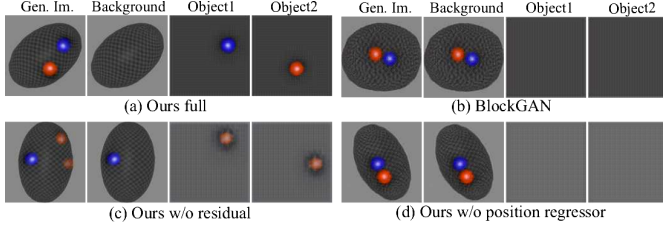

(a) Ours full        (b) BlockGAN

(c) Ours w/o residual        (d) Ours w/o position regressor

**Figure 2: Ablation Study.** For every case we render every component of our method independently. We show that only our full model is able to correctly disentangle individual components of the scene.

**Table 1: Ablation study.** FID score (lower is better) on BALLSINBOWL. Ours (full) reaches the highest fidelity by a large margin.

| | |
|---|---|
| BlockGAN 2D[29] | 152.3 |
| Ours w/o residual | 133.9 |
| Ours w/o pos. reg. | 154.8 |
| Ours (full) | **81.9** |

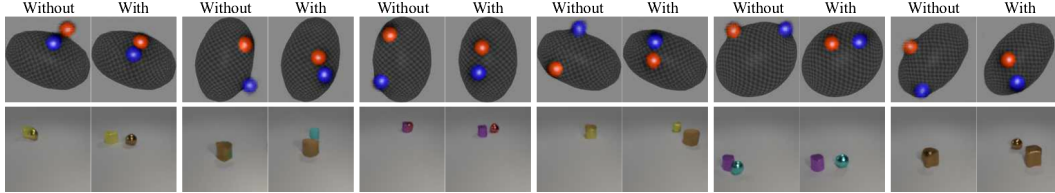

**Figure 3: Effect of the interaction module $\Gamma$.** We show pairs of images without and with the correction function $\Gamma$ is applied. For BALLSINBOWL the correction moves the balls within the bowl, and for CLEVR it pushes apart intersecting objects.

value $V_k(0) = e_v(z_k, z_0, \theta_k(0))$ is initialized as a function of the appearance parameters and initial positions (table A8); and we use the NPE style update equations [4], where $e_v$, $f_v$ and $g_v$ are MLPs,

$$v_k(t+1) = f_v(\theta_k(t), z_k, V_k(t), z_0, h_k^d(t)), \quad h_k^d(t) = \sum_{q \neq k} g_v(\theta_k(t), z_k, V_k(t), \theta_q(t), z_q, V_q(t)). \quad (4)$$

### 3.3 Learning objective

Training our model makes use of a training set $I_i$, $i = 1, \ldots, N$ of $N$ images of scenes containing different object configurations. No other supervision is required. Our learning objective is a sum of *two high fidelity losses* and *a structural loss* which we describe below.

For high fidelity, images $\hat{I}$ generated by the model above are contrasted to real images $I$ from the training set using the standard GAN discriminator $\mathcal{L}_{\text{GAN}}(\hat{I}, I)$ and style $\mathcal{L}_{\text{style}}(\hat{I}, I)$ losses from [29] (see section A3).

In addition, we introduce a regularizer to encourage the model to learn a non-trivial relationship between object positions and generated images. For this, we train a position regressor network $P$ that, given a generated image $\hat{I}$, predicts the location of the objects in it. In practice, we simplify this task and generate an image $\hat{I}'$ by retaining only object $k$ of the $K$ objects at random and minimizing $\|\breve{\theta}_k - P(G(W(z_0, z_k, \theta_k)))\|_2^2$. Here the symbol $\breve{\ }$ means that gradients are not back-propagated through $\theta_k$: this is to avoid mode collapse of the position at zero. $P$ shares most of its weights with the discriminator network (see table A11).

In the case of dynamic prediction, the discriminator takes as input the sequence of images concatenated along the RGB dimension and is tasked to discriminate between fake and real sequences. Similar to a static model we also have a position regressor which is tasked to predict the position of an object rendered at random with zero velocity.

## 4 Experiments

**Implementation details.** We learn mappings $\Psi_b$ and $\Psi_f$ using the same Adaptive Instance Normalization (AdaIN) [14] architecture. The spatial size of their output tensors is set to $H = 16$ and the final output image to $128 \times 128$ (which is reduced when needed for fair comparison to other methods). We used the Adam [18] optimizer for learning and train for a fixed number of epochs and always select the last model snapshot. We consider two types of baselines: standard generative models

**Table 2: Comparison to state-of-the-art methods.** FID score (lower is better) for various datasets. We consistently outperfom prior art in object centric scene generation. 'Ordered' refers to the variant discussed in sec. 3.2. †are standard GANs which FID score are evaluated on $64 \times 64$ images for 'General' variant (see section A4).‡is the 2D variant of BlockGAN [29] which sometimes fails to be object-centric (see section A1).

| RELATE variant | CLEVR-5 General | CLEVR-5vbg General | CLEVR General | ShapeStacks Ordered | REALTRAFFIC General |
|---|---|---|---|---|---|
| DCGAN† | 264.8 | 361.8 | 247.8 | 197.6 | 47.6 |
| DRAGAN† | 80.8 | 84.4 | 108.0 | **57.2** | **38.8** |
| OCF [1] | N/A | 83.1 | N/A | N/A | N/A |
| GENESIS [7] | 211.7 | 169.4 | 151.3 | 233.0 | 167.1 |
| BlockGAN2D ‡ [29] | 63.0 | 53.3 | 78.1 | 99.3 | 57.9 |
| Ours | **58.4** | **36.4** | **62.9** | 95.8 | 42.0 |

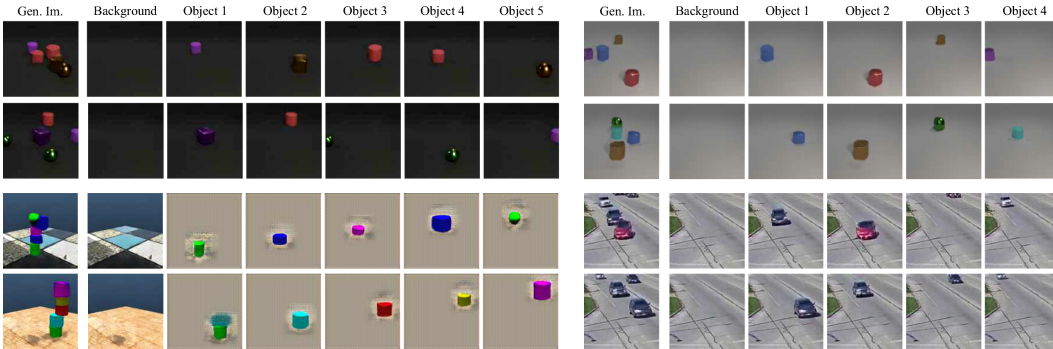

**Figure 4: Component-wise scene generation.** From a generated image (left) RELATE can render each component individually for each dataset. For CLEVR and REALTRAFFIC objects are rendered after being composed with the background (cf. section 4.2). Top left picture has increased contrast for easier visualization.

such as DCGAN[30] and DRAGAN[20], and object-centric generative baselines GENESIS [7] and OCF [1], quoting results from the original papers whenever possible. In addition we also added BlockGAN2D as an ablation of our method.

**Datasets.** We conduct experiments on four different datasets. First, we consider a relatively simple dataset, BALLSINBOWL from [6], for assessing the model features and ablations. This data consists of videos of two distinctly colored balls rolling in an elliptical bowl of variable orientation and eccentricity. Interactions amount to object collisions and the fact that they must roll within the bowl. To this, we add two popular synthetic datasets CLEVR [16] (cluttered tabletops) and ShapeStacks [12] (block stacking). Finally, we collected a new dataset REALTRAFFIC containing five hours of footage of a busy street intersection, divided into fragments containing from one to six cars. Especially the last dataset contains many interactions between the individual cars as they adapt their speed to the surrounding traffic which happens frequently when the light changes and cars either slow down because of a queue on red or accelerate when the lights change to green again. Further details about training, evaluation and datasets can be found in the appendix sections A4, A5 and A6.

## 4.1 Generating static scenes

**Ablation study.** We start experimenting with the comparatively simple BALLSINBOWL dataset to conduct basic ablations. The first ablation removes the spatial correlation module $\Gamma$ and the position regression loss, therefore reducing RELATE to a 2D version of BlockGAN. We also consider '*w/o residual*', where the addition $\hat{\theta}_k$ in equation (1) is removed, and '*w/o pos. loss*', where the position regression loss regularizer is removed. Table 1 shows that each component of RELATE yields an improvement in terms of FID scores on this dataset supporting our spatial modeling decisions. Furthermore, in figure 2 we show qualitatively that only RELATE (a) is able to correctly disentangle the underlying scene factors. We do this by generating the same image while retaining a single factor, which correctly isolates the background, and, in turn, both individual objects. BlockGAN 2D (b) and '*ours w/o pos. reg.*' (d) fail to disentangle the factors entirely, mapping everything to the

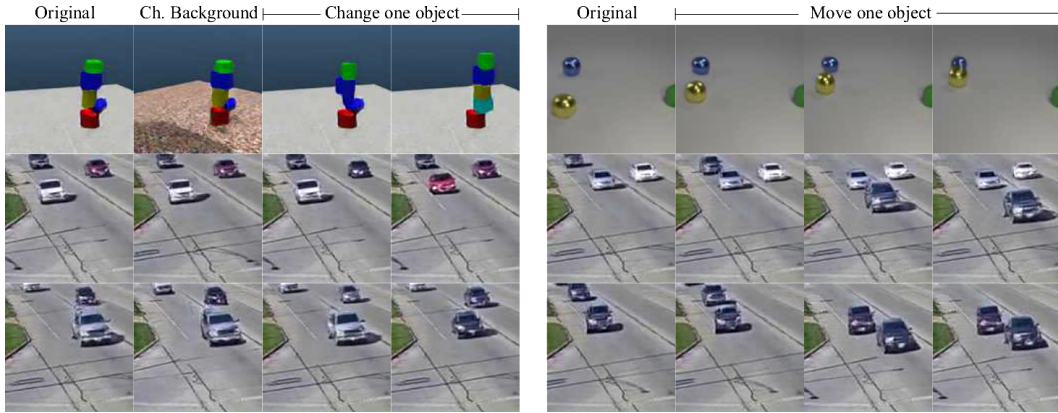

**Figure 5: Image editing.** Left: We demonstrate the capacity of RELATE to change the background and the appearance of individual objects. Right: RELATE is also able to modify the position of a single object.

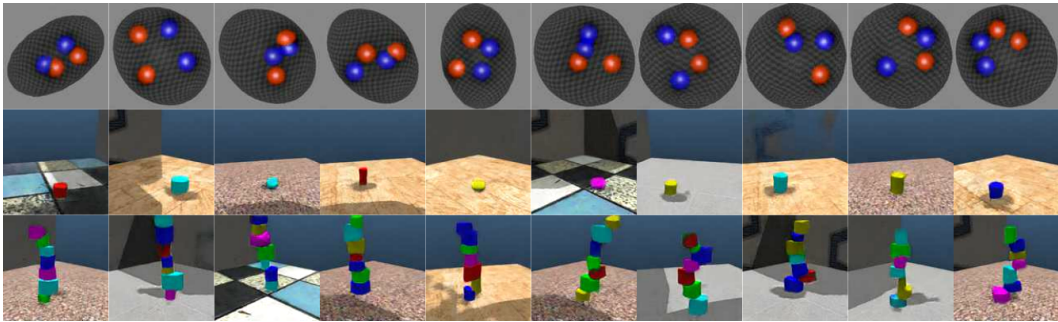

**Figure 6: Out-of distribution generation.** RELATE can generate images outside the training distribution. The first row shows generating a bowl with four balls, whereas the training set only features exactly two. The last two rows depict towers of one and seven objects, whereas the training images only had stacks of height two to five.

background component. *'Ours w/o residual'* (c) shows that the model partially fails to disentangle, with the background encoding some but not all the objects. Finally, figure 3 visualizes the effect of the interaction module $\Gamma$. Recall that this is implemented as a 'correction' function that accounts for correlation starting from independently-sampled parameters. For BALLSINBOWL, the correction module moves the balls within the bowl, and for the CLEVR it pushes objects apart if they intersect.

**Quantitative evaluation.** In table 2, we compare RELATE to existing scene generators on ShapeStacks, CLEVR and REALTRAFFIC. We report performance in terms of FID [13] score between 10,000 images sampled from our model and the respective test sets. For CLEVR, we train RELATE and BlockGAN on a restricted version of the data containing from three to six objects in an image[2], and at test time, we require all models to sample images with three to ten objects. We consistently outperform all prior object-centric methods in all scenes and scenarios according to FID scores. In particular, on CLEVR, RELATE can generate a larger number of objects than seen during training suggesting its improved generalization capabilities which are demonstrated further in figure 6. Our method also consistently out-performs standard GANs on all CLEVR datasets and is at par with DRAGAN on REALTRAFFIC. More qualitative results can be found in section A7.

## 4.2 Interpretability of the latent space and scene editing

As shown in the ablation studies in figure 2, RELATE successfully disentangles a scene into independent components - in contrast to BlockGAN2D which struggles to separate individual objects from the background. Figure 4 shows that RELATE can *disentangle* also far more complex scenes in REALTRAFFIC, ShapeStacks and CLEVR. Note that for REALTRAFFIC and CLEVR we render objects composed with the background. In fact in these datasets the size and appearance of each

Time

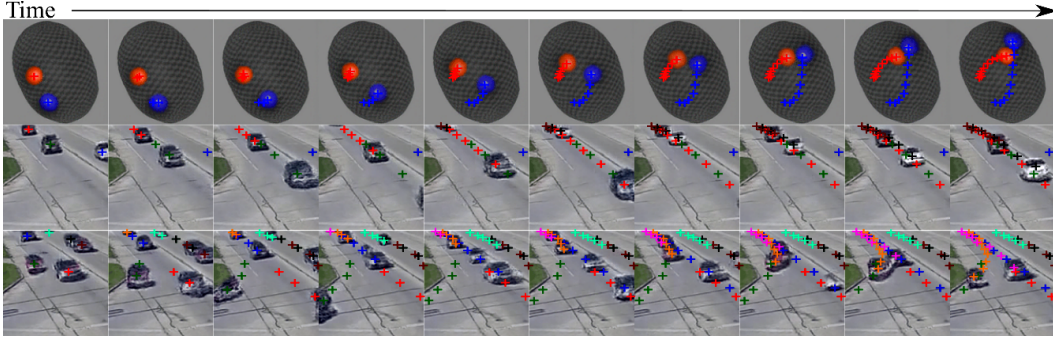

**Figure 7: Video generation.** We show consecutive video frames generated by RELATE overlayed with crosses representing projections of the model's estimated pose parameters for each object. In BALLSINBOWL the interaction with the environment is well captured as the balls stay within the bowl. In REALTRAFFIC the cars stay in their lane, or can decide to make a right turn (last row).

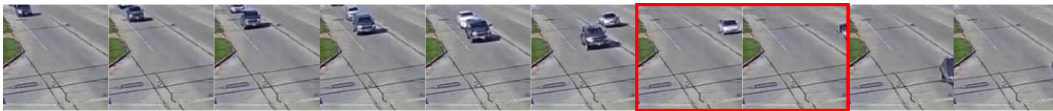

**Figure 8: Failure case.** Our model struggles to understand big changes of aspect ratio. For instance, in the static case, when we drag two cars towards the bottom of the image we can see that the left grey car disappears at some points (outlined in red) and reappears on the edge of the image. This explains why the video generation on REALTRAFFIC fails to capture the true data distribution more faithfully resulting in lower FVD score.

object is correlated to their position in the background because of the camera perspective. In addition to qualitative results, we also computed a disentanglement score in table A1 which measures how well our model is disentangling individual components of the scene. We found that our model manages to consistently separate each individual objects of the scene and outperform BlockGAN2D on the most challenging datasets which is inline with the qualitative evidence we observe. Next, in figure 5 we use RELATE to *edit* a generated scene. For example we can change the position or identity of individual objects. Finally, we show that RELATE can generate *out-of-distribution* scenes. This is achieved in particular by sampling a different number of objects. In figure 6, for instance, RELATE is trained on ShapeStacks seeing towers of height two to five. However, it can render taller towers of up to seven objects, or even just a single object. Likewise, in BALLSINBOWL it can generate bowls with four balls having seen only two during training. Furthermore, in ShapeStacks each tower is composed of blocks of *different* colors, but RELATE can relax this constraint rendering objects with repeated colors.

### 4.3 Simulating dynamics

We train this model on BALLSINBOWL and REALTRAFFIC to predict 15 and 10 consecutive frames respectively. During generation, we sample videos with a sequence length of 30 frames and measure the faithfulness with respect to the distribution of the test data via the *Fréchet Video Distance (FVD)* [34]. We achieve FVD scores of 555.5 and 2252.7 respectively. This is perceptibly better than 919.9 and 3369.6 for a baseline consisting of time-shuffled sequences from the respective training sets, which feature perfect resolution but poor dynamics. Qualitatively in figure 7 we see that the model does understand the motion and captures interaction with the background. For instance in BALLSINBOWL the balls do have a curved motion because of the shape of the bowl and decrease in speed when reaching the edges of the bowl which are in higher position (see first row of figure 7). In REALTRAFFIC the cars do stay in their respective lane. Interestingly our model is able to handle different types of motions correctly (see third row figure 7) and use the sample vector to decide whether the cars should go straight or turn. Finally we see that we can also generate videos with much more cars than the upper bound (5) with which the system was trained (see last row figure 7).

**Limitations** While the dynamics in video generation look realistic in most cases, REALTRAFFIC also exposed the limitations of our approach. In this dataset the perspective range of the camera is important. As a result, cars at the bottom of the image, which appear bigger in the training data, often

do not get generated properly in the static case. We hypothesize that this is also the main reason why the cars' appearance (and hence FVD score) deteriorates in the dynamic scenarios: since the style parameters $z_i$ are fixed to preserve identity, it is not possible for the model to account accurately for the appearance change introduced by large changes of perspective over the course of a sequence.

## 5  Conclusion

We have introduced RELATE, a GAN-based model for object-centric scene synthesis featuring a physically interpretable parameter space and explicit modeling of spatial correlations. Our experimental results suggest that spatial correlation modeling plays a pivotal role in disentangling the constituent components of a visual scene. Once trained, RELATE's interpretable latent space can be leveraged for targeted scene editing such as altering object positions and appearances, replacing the background or even inserting novel objects. We demonstrate our model's effectiveness by presenting *state-of-the-art* scene generation results across a variety of simulated and real datasets. Lastly, we show how our model naturally extends to the generation of dynamic scenes being able to generate entire videos from scratch. A main limitation of our current model is its restriction to planar motions which prevents it from representing arbitrarily 3D motions featuring angular rotation more faithfully, most notably highlighted by the experiments for video generation. However, we believe that our work can contribute to future research in that area by providing a scalable spatio-temporal modeling approach which is conveniently trainable on unlabeled data.

## Broader Impact

Our method advances the ability of computers to learn to understand environments in images in an object-centric way. It also enhances the capabilities of generative models to generate realistic images of "invented" environment configurations.

Overall, we believe our research to be at low to no risk of direct misuse. At present, our generation results are insufficient to fool a human observer. However, it has to be noted that the sampling process is, as in many other deep generative models, capable of revealing patterns observed in the training data, *e.g.*, specific textures or object geometries. Such data privacy concerns are not applicable in the street traffic data used in our research, since the resolution of the videos is far too low to identify individual drivers or recognize cars' license plates. However, 'training data leakage' should be taken into consideration when the model is trained on more sensitive datasets.

In a positive prospect, we believe that our model contributes to further the development of less opaque machine learning models. The explicit object-centric modelling of image components and their geometric relationships is in many of its aspects intelligble to a human user. This facilitates debugging and interpreting the model's behaviour and can help to establish trust towards the model when employed in larger application pipelines.

However, the key value of our paper is in the methodological advances. It is conceivable that, like any advance in machine learning, our contributions could ultimately lead to methods that in turn can and are misused. However, there is nothing to indicate that our contributions facilitate misuse in any direct way; in particular, they seem extremely unlikely to be misused directly.

## Acknowledgments

This work is supported by the European Research Council under grants ERC 638009-IDIU, ERC 677195-IDIU, and ERC 335373. The authors acknowledge the use of Hartree Centre resources in this work. The STFC Hartree Centre is a research collaboratory in association with IBM providing High Performance Computing platforms funded by the UK's investment in e-Infrastructure. The authors also acknowledge the use of the University of Oxford Advanced Research Computing (ARC) facility in carrying out this work (`http://dx.doi.org/10.5281/zenodo.22558`). The authors also thank Olivia Wiles for feedback on the paper draft, Thu Nguyen-Phuoc for providing the implementation of BlockGAN and Titas Anciukevičius for providing the generation code for CLEVR variants. We finally thank anonymous reviewers for precious feedback.

## Footnotes

*indicates equal contribution

[2]Note that GENESIS was trained on the full training set featuring three to ten objects.

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
