[Supplementary Material · relate_supplementary.pdf]

# Supplementary material for "RELATE: Physically plausible Multi-Object Scene Synthesis Using Structured Latent Spaces"

In this supplementary material we provide further details about RELATE. This manuscript is organised as follow. We first give additional object decomposition score in section A1. Further discussions about the method are presented in section A2. We describe the losses we use in more details in section A3. We follow with a more in-depth description of implementation details in section A4. Section A5 is dedicated to explain in more details how the various baselines were trained. Section A6 contains a more thorough explanation of every dataset and data collection when applicable. Finally we provide more qualitative results in section A7.

## A1 Additional experiments

**Table A1: Disentanglement score.** For each dataset of table 2 we report respectively the distance and correlation score described in section A1. Our model outperforms BlockGAN2D in the most complex scenario: ShapeStacks and BALLSINBOWL. Both model reach similar scores for the other scenes.

| | CLEVR-5 General | CLEVR-5vbg General | CLEVR General | ShapeStacks Ordered | REALTRAFFIC General | BALLSINBOWL General |
|---|---|---|---|---|---|---|
| BlockGAN2D | 19.0 | 18.0 | 18.0 | 272.0 | 22.0 | 98.0 |
| Ours | 17.0 | 17.0 | 19.0 | 17.0 | 23.0 | 26.0 |

To provide more quantitative insights of the disentanglement capabilities of our model we ran additional experiments. While measures such as MIG [26] can be used, computing this score is not applicable in our case since our model does not feature an inference component to compute the posterior $q(z|x)$. As an alternative, we toggle each object individually (out of 5 objects generated), looking at how the generated image changes. We then report the distance between the pixel location corresponding to the maximum image change and the location (scaled $\theta_i$) of the object that was toggled. In table A1 we report the median distance between $\theta_i$ and the pixel location corresponding to the maximum image change. We note that in general that our model outperforms BlockGAN2D, most notably for ShapeStacks where BlockGAN isn't disentangling different objects at all. In addition we note that for the model trained on ShapeStacks, the discriminator can predict bottom object position with 11.3 mean pixel errors on the test set.

## A2 Further Discussions

As noted in the conclusion, this paper only tackles 2D representations. However our model is generic enough to be exported to 3D and could be for instance plugged directly into BlockGAN[29]. This would result in longer training time. Besides one would need to adapt the architecture so as to generate image with similar size as this paper (our generated images are twice as large as BlockGAN[29] ones). Finally while this work is limited to object represented by centroid positions, another possible extension could be to combine it with [1] in order to also provide segmentation maps for each objects.

## A3 Losses

Our final loss is the sum of three losses mentioned in the main text:
$$\mathcal{L}_{tot} = \mathcal{L}_{\text{GAN}}(\hat{I}, I) + \mathcal{L}_{\text{style}}(\hat{I}, I) + \min_{G,\Gamma,P} \|\check{\theta}_k - P(G(W(z_0, z_k, \theta_k)))\|_2^2,$$

where $\mathcal{L}_{\text{GAN}}(\hat{I}, I)$ is the standard GAN loss:
$$\mathcal{L}_{\text{GAN}}(\hat{I}, I) = \min_{G,\Gamma} \max_D \mathbb{E}[\log(1 - D(G(W(Z, \Theta))))] + \mathbb{E}[\log(D(I))]$$

The style loss follows the implementation of BlockGAN [29]. The input of the style discriminator $D_l$ are mean $\mu_l$ and variance $\sigma_l^2$ across spatial dimensions of $\Phi_l(x) \in \mathcal{R}^{W_l \times H_l \times C_l}$, the output of the $l$th layer of $D$ taken before the normalization step:
$$\mu_l(\Phi_l(x)) = \frac{1}{W_l \times H_l} \sum_i \sum_j \Phi_l(x)_{i,j},$$

**Table A2: Hyper-parameters for each datasets.** Epoch nums are the number of epochs we trained for.

| Dataset | Learning rate | Epoch nums | $M$ | $K_{min} - K_{max}$ | $H'$ | $N_b$ | $N_f$ | $H''/H$ sampling range |
|---|---|---|---|---|---|---|---|---|
| BALLSINBOWL | 0.001 | 60 | 1 | 2-2 | 8 | 3 | 1 | $[-0.8, 0.8]^2$ |
| CLEVR5 | 0.0001 | 40 | 2 | 2-5 | 4 | 1 | 90 | $[-0.6, 0.6]^2$ |
| CLEVR5-vbg | 0.0001 | 30 | 2 | 2-5 | 4 | 1 | 90 | $[-0.6, 0.6]^2$ |
| CLEVR | 0.0001 | 40 | 2 | 3-6 | 4 | 1 | 90 | $[-0.6, 0.6]^2$ |
| ShapeStacks | 0.001 | 30 | 2 | 2-5 | 4 | 12 | 64 | $[-0.6, 0.6] \times [0, 0.6]$ |
| REALTRAFFIC | 0.0001 | 20 | 2 | 1-5 | 6 | 1 | 20 | $[-0.6, 0.6]^2$ |

$$\sigma_l^2(\Phi_l(x)) = \frac{1}{W_l \times H_l} \sum_i \sum_j (\Phi_l(x)_{i,j} - \mu_l(\Phi_l(x)))^2.$$

The style discriminator $D_l$ for each layer is then implemented as a linear layer followed by a sigmoid activation function. The resulting style loss is:

$$\mathcal{L}_{\text{style}}(\hat{I}, I) = \max_D \sum_l \mathbb{E}[\log(1 - D_l(\hat{I}))] + \mathbb{E}[\log(D_l(I))]$$

## A4  Implementation details

**Infrastructure and framework**   For all experiments we use PyTorch 1.4. We train all models on a single NVIDIA Tesla V100 GPU.

**Training hyperparameters**   We initialize all weights (including instance normalization ones) by drawing from a random normal distribution $\mathcal{N}(0, 0.02)$. All biases were initialized to 0. For each update of the discriminator we update the generator $M$ number of times. We use Adam parameters $(\beta_1, \beta_2) = (0., 0.999)$ for all datasets except BALLSINBOWL where $\beta_1 = 0.5$. Similarly $W$ was a max-pooling operator in all datasets except BALLSINBOWL where we used a sum pooling operator. As in BlockGAN[29], background and foreground decoders each start from a learned constant tensors $T_b$ and $T_f$ respectively with sizes $H \times H \times 256$ and $H' \times H' \times 512$. For BALLSINBOWL we use a tensor $T_f$ for each object and use a constant style vector of one.

In the case of dynamic scenarios we reuse same hyperparameters as in the static case except that we use a learning rate of 0.0001 and $\beta_1 = 0$.

Full details of the parameters for each dataset can be found in table A2

**Evaluation details**   For FID scores computation we draw $10\,000$ samples from our model which we compare against the same number of images drawn from the test set. To compute FVD score on each dataset, we sample 500 videos of 30 frames from our model and compare them against the videos of the respective test sets (500 for BALLSINBOWL and 275 videos for REALTRAFFIC). This also applies to the time shuffled baseline.

To be able to compare with other methods we resize our generated images to $96 \times 96$ on CLEVR5 and CLEVR5-vbg and $64 \times 64$ for ShapeStacks. For the simple generative baselines, DRAGAN and DCGAN we evaluate FID score on the generated $64 \times 64$ images from these models. We evaluate on the generated $128 \times 128$ images otherwise.

We empirically found that background was rendered with better quality for lower values of $z_0$. Hence at test time we sampled $z_0$ from $\mathcal{U}([-0.5, 0.5]^{N_b})$ for optimal results.

### A4.1  Architecture details

**Generator.**   In this work we maintain the core of our architecture fixed as much as possible. Since the dimension of the sample $z_i$ does not necessarily match the channel dimension where it is injected before applying Adaptive Instance Normalisation (AdaIN) to a layer $l$ we map $z_i$ to a vector $\hat{z}_i$ transformed such that

$$\hat{z}_i = \max(W_l^T z_i + b_l, 0)$$

Where $(W_l, b_l)$ are learnable parameters. AdaIN is applied at the end of the layers (after the activation). All LeakyReLU layers are using a parameter of 0.2.

**Table A3: Network architecture for the foreground object generator $\Psi_f$.**

| Layer name | Layer Type | Input size | Output size | Kernel Size | Stride | Activation | Norm. |
|---|---|---|---|---|---|---|---|
| Style_f | Id | $H' \times H' \times 512$ | $H' \times H' \times 512$ | - | - | Id | AdaIn |
| Convtf_1 | ConvTranspose | $H' \times H' \times 512$ | $H' \times H' \times 512$ | $3 \times 3$ | 1 | LeakyReLU | AdaIn |
| Convtf_2 | ConvTranspose | $H' \times H' \times 512$ | $H' \times H' \times 256$ | $3 \times 3$ | 1 | LeakyReLU | AdaIn |
| Pad | Padding | $H' \times H' \times 512$ | $H \times H \times 256$ | - | - | - | - |

**Table A4: Network architecture for the background object generator $\Psi_b$.**

| Layer name | Layer Type | Input size | Output size | Kernel Size | Stride | Activation | Norm. |
|---|---|---|---|---|---|---|---|
| Style_b | Id | $H \times H \times 256$ | $H \times H \times 512$ | - | - | Id | AdaIn |
| Convtb_1 | ConvTranspose | $H \times H \times 512$ | $H \times H \times 512$ | $3 \times 3$ | 1 | LeakyReLU | AdaIn |
| Convtb_2 | ConvTranspose | $H \times H \times 512$ | $H \times H \times 256$ | $3 \times 3$ | 1 | LeakyReLU | AdaIn |

**Table A9: Network architecture for module $f_0$.**

| Layer name | Layer Type | Input size | Output size | Activation |
|---|---|---|---|---|
| FC$f_0\_1$ | Linear | $N_f + N_b + 2$ | 128 | LeakyReLU |
| FC$f_0\_2$ | Linear | 128 | 64 | LeakyReLU |
| FC$f_0\_3$ | Linear | 64 | 2 | Tanh |

**Table A10: Network architecture for module $f_1$.**

| Layer name | Layer Type | Input size | Output size | Activation |
|---|---|---|---|---|
| FC$f_1\_1$ | Linear | $N_f + N_b$ | 128 | LeakyReLU |
| FC$f_1\_2$ | Linear | 128 | 64 | LeakyReLU |
| FC$f_1\_3$ | Linear | 64 | 2 | None |
| $Pos_{out}$ | Sigmoid(x) Tanh(y) | 2 | 2 | None |

**Discriminator** We describe the architecture of the discriminator network in more details in table A11. We use spectral normalization [27] at almost every layer. Positions are directly regressed from the last feature output of the discriminator (see last line $P_{end}$). Therefore in practice $P$ and $D$ share the same backbone $D_b$ (see table table A11 until flatten) for every image I:

$$P(I) = P_{end}(D_b(I)), \qquad D(I) = Disc(D_b(I)).$$

Input for style discriminator are taken after the convolution of (Convd_2, Convd_3, Convd_4, Convd_5) in table A11 before the normalization. Spectral Normalization was *not* applied to any $D_l$.

## A5 Baselines

**DCGAN[30] and DRAGAN[20].** We used an online pytorch implementation[3] with default hyperparameters. We trained these models to generate $64 \times 64$ images and therefore only evaluated FID score at the same resolution (see section A4).

**OCF.** OCF results were copied from original paper of [1].

**BlockGAN2D.** We use the same hyperparameters and network architecture as RELATE except for learning rate and $M$. In all cases we report the best results over models trained with variations of learning rate in (0.001, 0.0001) and $M$ in (2,3).

**GENESIS.** We use the official implementation[4] of GENESIS for all experiments. For the ShapeStacks dataset, we use the official model snapshot released with the original paper[5]. For all other datasets, we train GENESIS for 500,000 iterations with the default learning parameters and select the last model checkpoint for evaluation. When training GENESIS we use *constrained ELBO optimization* [31] controlled via g_goal in the training script which influences the decomposition capability of GENESIS. We perform a grid search over g_goal in the range of 0.5635 to 0.5655 and select the model with the lowest ELBO after 500,000 iterations.

**Table A5: Network architecture for the generator G.** Outputs of all $K$ foreground object generators $\Psi_f$ and background generator $\Psi_b$ are stacked on the first dimension before entering layer $W$ (third row).

| Layer name | Layer Type | Input size | Output size | Kernel Size | Stride | Activation |
|---|---|---|---|---|---|---|
| $\Psi_f$ (see table A3) | - | $H' \times H' \times 512$ | $16 \times 16 \times 256$ | - | - | - |
| $\Psi_b$ (see table A4) | - | $H \times H \times 512$ | $16 \times 16 \times 256$ | - | - | - |
| W | Max/Sum Pool | $(K+1) \times 16 \times 16 \times 256$ | $16 \times 16 \times 256$ | - | - | - |
| Convtg_1 | ConvTranspose | $16 \times 16 \times 256$ | $32 \times 32 \times 128$ | $4 \times 4$ | 2 | LeakyReLU |
| Convtg_2 | ConvTranspose | $32 \times 32 \times 128$ | $64 \times 64 \times 64$ | $4 \times 4$ | 2 | LeakyReLU |
| Convtg_3 | ConvTranspose | $64 \times 64 \times 64$ | $64 \times 64 \times 64$ | $3 \times 3$ | 1 | LeakyReLU |
| Convtg_4 | ConvTranspose | $64 \times 64 \times 64$ | $128 \times 128 \times 64$ | $4 \times 4$ | 2 | LeakyReLU |
| Convtg_5 | ConvTranspose | $128 \times 128 \times 64$ | $128 \times 128 \times 3$ | $3 \times 3$ | 1 | Tanh |

**Table A6: Network architecture for module $f$ and $f_v$.** * indicates modification of $f_v$

| Layer name | Layer Type | Input size | Output size | Activation |
|---|---|---|---|---|
| FC$f$_1 | Linear | $2 \times (N_f + 2 + 2^*)$ | 32 | LeakyReLU |
| FC$f$_2 | Linear | 32 | 32 | LeakyReLU |
| FC$f$_3 | Linear | 32 | 32 | None |

## A6  Datasets

**BALLSINBOWL.**   This dataset is a replica of the two balls synthetic dataset of [6]. It consists of 2500 training sequences and 500 test sequences of two balls of different fixed colour rolling in bowls of various shapes. We count an epoch as 10,000 iterations over the data. In figure A1 we show some sample data from this dataset.

**CLEVR.**   We used the official CLEVR from [16]. We train on data from train and validation set and evaluate on the test set. Both ours and BlockGAN2D were trained on the subset containing 3 to 6 objects and evaluated on the entire test set.

**CLEVR5/CLEVR5-vbg.**   We use online code provided by the authors[6] to generate CLEVR5 and CLEVR5-vbg. As done in [1] we generate 100,000 images keep 90,000 for training and 10,000 for testing.

**ShapeStacks**   We use the official release of the ShapeStacks dataset[7]. We use the `default` partitioning provided with the dataset and merge the training and validation splits for a total of 264,384 training images. All FID comparisons are made against 10,000 images randomly sampled from the test set which contains 46,560 images in total. Since the original resolution of the images is $224 \times 224$ pixels, we re-scale them to $128 \times 128$ before feeding them to our network.

**REALTRAFFIC.**   We recorded 5 hours from Youtube[8] of a live traffic camera at a crossing. The video was then unrolled at 10 fps and manually processed to keep only sequences with a number of cars in [1,5]. We kept 560 videos for the training set and 123 in test (80/20 ratio). This dataset will be publicly released.

## A7  Qualitative results

We provide additional qualitative generation results. Figure A2 shows a failure case of BlockGAN2D mentionned in the paper. In fact, when the scene is more structured BlockGAN2D fails to be object centric and let the background render the entire scene. In addition figures A3, A5, A6, A7 and A8 provide more samples on every dataset for all the models we trained. In particular we can see that when inter-objects relations are weak in CLEVR5 or CLEVR5-vbg, BlockGAN2D performs qualitatively similar to ours (see figures A5 and A6). However when the scene is more crowded and

**Table A7: Network architecture for module $g$ and $g_v$.** $^*$ indicates modification of $g_v$

| Layer name | Layer Type | Input size | Output size | Activation |
|---|---|---|---|---|
| FCg_1 | Linear | $32 + N_f + 2 + 2^* + N_b$ | 32 | LeakyReLU |
| FCg_2 | Linear | 32 | 32 | LeakyReLU |
| FCg_3 | Linear | 32 | 2 | Tanh |

**Table A8: Network architecture for module $e_v$.**

| Layer name | Layer Type | Input size | Output size | Activation |
|---|---|---|---|---|
| FC$e_v$_1 | Linear | $N_f + 2 + N_b$ | 128 | LeakyReLU |
| FC$e_v$_2 | Linear | 128 | 128 | LeakyReLU |
| FC$e_v$_3 | Linear | 128 | $3 \times 2$ | Tanh |

the objects have higher correlation BlockGAN2D quality decreseases significantly (see figures A3, A7 and A8).

**Table A11: Network architecture for the discriminators.** Note that the Instance Normalization weights were also subjected to spectral normalization. $P$ and $D$ shares weights until Flatten layer.

| Layer name | Layer Type | Input size | Output size | Kernel Size | Stride | Activation | Norm. |
|---|---|---|---|---|---|---|---|
| Convd_1 | Conv | $128 \times 128 \times 3$ | $64 \times 64 \times 64$ | $5 \times 5$ | 2 | LeakyReLU | - |
| Convd_2 | Conv | $64 \times 64 \times 64$ | $32 \times 32 \times 128$ | $5 \times 5$ | 2 | LeakyReLU | IN/Spec. Norm. |
| Convd_3 | Conv | $32 \times 32 \times 128$ | $16 \times 16 \times 256$ | $5 \times 5$ | 2 | LeakyReLU | IN/Spec. Norm. |
| Convd_4 | Conv | $16 \times 16 \times 256$ | $8 \times 8 \times 512$ | $5 \times 5$ | 2 | LeakyReLU | IN/Spec. Norm. |
| Convd_5 | Conv | $8 \times 8 \times 512$ | $4 \times 4 \times 1024$ | $5 \times 5$ | 2 | LeakyReLU | IN/Spec. Norm. |
| Flatten | Id | $4 \times 4 \times 1024$ | $1 \times 1 \times 16384$ | - | - | - | - |
| $Disc$ | Linear | $1 \times 1 \times 16384$ | 1 | - | - | Sigmoid | None/Spec. Norm. |
| $P_{end}$ | Linear | $1 \times 1 \times 16384$ | 2 | - | - | Tanh | None/Spec. Norm. |

**Figure A1: Sample data from BALLSINBOWL.** The dataset consists of two balls of different colors rolling in elliptical bowls of various shapes.

**Figure A2: BlockGAN2D scene decomposition on ShapeStacks.** We display in order (generated, background, object_1, ..., object_5) for BlockGAN2D model. We see that in this case BlockGAN doesn't capture objectness at all and render everything in the background. This shows how, for structured scenes, prior work fails to capture correlations between objects.

RELATE

BLOCKGAN2D

**Figure A3: Generated scenes for models trained on BALLSINBOWL.** Qualitatively RELATE generates images of higher quality compared to BlockGAN2D[29].

RELATE

BLOCKGAN2D

GENESIS

DCGAN

DRAGAN

**Figure A4: Generated scenes for models trained on ShapeStacks.** Despite qualitative similar rendering, BlockGAN2D isn't rendering a scene component-wise as opposed to ours (see figure A2).

RELATE

BLOCKGAN2D

GENESIS

DCGAN

DRAGAN

**Figure A5: Generated scenes for models trained on CLEVR5.** For less crowded scenes our model and BlockGAN2D reach similar performances.

RELATE

BLOCKGAN2D

GENESIS

DCGAN

DRAGAN

**Figure A6: Generated scenes for models trained on CLEVR5-vbg.** This scenario reaches similar conclusion as figure A5.

RELATE

BLOCKGAN2D

GENESIS

DCGAN

DRAGAN

**Figure A7: Generated scenes for models trained on CLEVR.** When the scene gets more crowded RELATE gets an advantage as it can push objects apart resulting in higher qualitative rendering.

**Figure A8: Generated scenes for models trained on REALTRAFFIC.** Our model qualitatively renders higher fidelity images. BlockGAN2D sometimes suffers from background mode collapse (see first, second and fourth rows of second block).

## Footnotes

[3]`https://github.com/LynnHo/DCGAN-LSGAN-WGAN-GP-DRAGAN-Pytorch`

[4]`https://github.com/applied-ai-lab/genesis`

[5]`https://drive.google.com/drive/folders/1uLSV5eV6Iv4BYIyhOR9DUGJT2W6QPDkb?usp=sharing`

[6] https://github.com/TitasAnciukevicius/clevr-dataset-gen.

[7] https://shapestacks.robots.ox.ac.uk/#data

[8] https://www.youtube.com/watch?v=5_XSYlAfJZM