[Reviews · NeurIPS 2020]

Review 1

Summary and Contributions: The paper presents a new generative model that focuses on composition and generates images while accounting for the multiple objects within them and their interaction. It shows experiments over synthetic data as well as natural albeit a bit limited natural data (photos of cars on the road from above). Since the scenes are generated starting from interpretable parameters for the objects positions, it allows for controlled generation. Finally the model can generate short videos that model the interaction between objects. Update: after I read the other reviews I actually find multiple concerns about the paper: I agree with the other reviewers about the issue of physical plausibility, the partial independence assumptions (the initial CNN layers are not shared and therefore some of the interactions cannot be captured by the model), and the potential limited novelty given BlockGAN. I therefore unfortunately feel I should change the score to 6 I sincerely apologize about it.

Strengths: A new model that explores compositionality in generative models. Learning about objects in an unsupervised manner. Scaling to short video generation. Generalization to out-of-distribution scenes (e.g. more/less objects than training).

Weaknesses: It would be useful if there were experiments for realistic images of higher complexity than the cars’ example, as they are quite clean and with relatively low diversity compared to most natural images. It looks like the model makes multiple assumptions the validity of which for natural images or most real scenes is unclear to me. One assumption they make is that the visual appearance of different object is independent (“Different from the object posed below, they are sampled independently, thus assuming that the appearance of different objects is independent.”) However, in many realistic scenes, there are relations in visual appearance between the different objects. That’s even the case in CLEVR with e.g. object reflections. Another assumption is that “individual objects are much smaller than the overall scene” which of course is not necessarily valid as well. I’m therefore wondering if this approach can apply to any realistic scenes beyond the particular tasks in the paper, that all share these assumptions. The assumption about a particular pose for each object also implicitly makes assumptions on the shape and rigidity of the objects. Will the approach apply to visual entities that span beyond a particular small blob in the picture that could be mostly accounted for by its centroid position?

Correctness: The claims in the paper are corroborated by experiments and the model description is clear and correct.

Clarity: The paper clarity is good and it is well written and easy to follow.

Relation to Prior Work: The related work section presents a good discussion of relations to other works and gives the right context for the paper.

Reproducibility: Yes

Additional Feedback: * The object images are merged together using max or sum-pooling of images for different components. In practice this may operate similarly to having different objects reside in different layers, where again, it makes it unclear to me whether this will allow for more varied visual interactions between objects than just occlusions. * For the correct module, it seems like it can make a correction only once given the initial sampled positions, rather than keep refining the new positions based on the other updated new positions. I believe running this layer over multiple iterations might help in that regard (e.g. if two object positions initially don’t intersect with each other but after the update due to interactions with other objects position they become interesting, and then require further position refinement). * Did you try predicting the positions of all the objects at once? Why did you choose predicting them per each separated image? * In figure 5, the model didn’t really change only one object. You can see some shapes change for the other objects as well. Why is that the case given the architecture of the model and could it be mitigated? * In line 207, it says that objects are being pushed apart if they intersect. However, in the CLEVR dataset occlusions of objects do exist. Can the model account for that, or is that the case that because it uses pooling the merge the images together, potentially making the merged image less realistic when objects overlap, the model learns to prefer to not make them overlap (a case which ideally we want the model to account for). * In line 218, I thought that the difference between BlockGAN and the new approach is the consideration of interactions between object positions. If so, do you have intuition as to why BlockGAN struggles to separate out objects from the background?


Review 2

Summary and Contributions: This paper proposes an object-aware GAN architecture which incorporates object correlations through graph neural networks on object latent representations. The model extends a 2D variant of BlockGAN with a graph neural network layer on the latent state to model object relations and dynamics. In experiments, improved image and video generation is demonstrated based on the FID score compared to previous works.

Strengths: * The paper proposes a novel combination of BlockGAN with graph neural networks on the object latents. * The approach seemingly improves image synthesis performance in terms of FID score.

Weaknesses: * The paper claims that it would show that modelling object correlations is necessary to disentangle position and identity. This is not appropriately shown by the paper. Only a single empirical qualitative example is given in Fig. 2 which could just be an artifact of the training. Disentanglement measures such as mutual information gap (MIG) should be used for quantitative evaluation, see for instance Locatello et al., On the Fairness of Disentangled Representations, https://arxiv.org/pdf/1905.13662.pdf. * The FID score gives the similarity in VGG image features between generated images and test set. It is difficult to interpret beyond textural similarity. It does not measure disentanglement of faithfulness of the reconstruction in the objects and their positions. * The ablation study in Tab 1, Fig. 2 is inconclusive on which model component could lead to disentanglement. It might also be an artifact of hyperparameters or the training runs. * l. 163, the position regularizer is not well explained. Please clarify what a trivial relationship between object position and generated image should be. This loss seems to force the network to learn to map \theta in such a way that it can be observed from the image. Why is the shift by theta of the feature map not sufficient? Please elaborate. * How robust is the training? Does it always converge to similar results? How does this affect the results of the variants in the ablation study and their comparability? The paper should report average results over multiple random training runs. * Why can BlockGAN not represent the different objects in the BallsInBowl data? * How accurate is the object position prediction by the network in BallsInBowl or ShapeStacks? Since the scenes are rendered synthetically and ground truth positions are available, evaluating position error could give a better hint on the quality of the position estimate than FID. * Fig. 7 does not show object collisions. How well can the model capture collision events? This is typically challenge to incorporate in a latent dynamics model. The paper should also demonstrate the approach on falling towers on ShapeStacks. * The approach does not have a physically interpretable latent state and does not learn the physics-based dynamics of the underlying scene. In the cars scene, the dominating factor of the dynamics seems to be traffic interaction. Hence the method learns some dynamics in some 2D projection. Writing on neural physics engines in intro, related work or method seems off topic.

Correctness: * The paper does not justify the claim that modelling object correlations is necessary to disentangle position and identity. See paper weaknesses. * The notion of "physical plausibility" is questionable. The architecture includes a graph NN layer which seems to allow the network to refine the scene representation based on object latent interactions. Actual physical plausibility such as reasoning on 3D scene configuration, respecting collisions or Newton's laws incl external forces like gravity are not incorporated into this model. The authors should revise and remove the claim of "physical plausibility".

Clarity: The paper is well written and clear.

Relation to Prior Work: The paper provides a discussion of prior work and explains the differences of the proposed model to previous works. It also provides comparisons with several state-of-the-art models.

Reproducibility: No

Additional Feedback: * l 196, eq 4 => eq 1?


Review 3

Summary and Contributions: The paper presents a new generative model (RELATE) for the synthesis of images of scenes with multiple objects from physically interpretable latent vectors. The general approach is similar to the BlockGAN and extends is with the following key contributions: * a correction network that models the interactions between the object in the scene * a structural loss function that evaluates the poses of objects in the scene

Strengths: * The discussed problem is relevant in the field of generative modelling for image synthesis * The presented approach extends the BlockGAN 2D baseline for dynamical image synthesis with 2D motion

Weaknesses: * The model leverages multiple assumptions that seems to be unfeasible in practice 1. line 99 -- It is assumed that scene consists of K objects. However, when training on unlabelled real images the number of objects in the scene is unknown. The BlockGAN baseline performs training on a single instance real images and then generalises to multi-instance images. How the K parameter is selected during the training? If the manual labelling of the average number of objects is required, then the method does require labeled data (as opposed to unlabelled data in line 3). If the K is chosen randomly, the discriminator will notice the difference between real and synthetic images. 2. line 109 -- For simplicity, we assume \theta_k in R^2 to be a 2D translation. The BlockGAN baseline uses the 3D poses to model 3D scenes. If the RELATE model operates only with 2D poses, it should fail to generate true perspective scenes (the example with the car traffic in the Supplementary is nearly 2D). A field of application of a RELATE that can model only 2D dynamics (e.g., top view) seems to be limited. * Some details regarding the preparation of the training dataset in terms of the number of objects (K) are required to reproduce the reported results.

Correctness: * The presented method is correct for generating still images and 2D scene dynamics. * Generalization to 3D dynamics scenarios and perspective scenes seems to be limited.

Clarity: * The paper is generally easy to follow and clearly written. * Still multiple typos are present, e.g., - line 94 -- RELATE (figure 1) -> RELATE (Figure 1), all references to figures, tables, etc. should start with a capital letter, e.g., Figure - line 107 -- it is better to notify the reader that the Table A1 is in the supplementary material * Some sentences are confusing, e.g., - line 100 -- individual foreground object -- How does network will distinguish foreground and background for complex scenes, e.g., car driving behind a tree?

Relation to Prior Work: * Authors use a BlockGAN model as a starting point for their research. * The Related Work section is well structured.

Reproducibility: No

Additional Feedback: * I would consider, changing the overall score, if the authors answer the following questions: 1. Table 1. The FID score measures the distance between real and synthetic images in static. If the BlockGAN and RELATE use the same generator network, why does dynamics modeling affect the FID distance? 2. line 99 – Is it possible to train the RELATE using image sequences from the GRAM dataset [1]? If yes, how the K is chosen? [1] http://agamenon.tsc.uah.es/Personales/rlopez/data/rtm/ # Additional feedback The authors have made good job in the rebuttal. Additional training results prove the strength of the proposed method.


Review 4

Summary and Contributions: The authors propose an object-oriented, GAN-based architecture which generates scenes by first generating object representations, and then composing those representations into an output image. It improves on past work by using a relational architecture for generating the objects, which allows the network to model dependencies between objects. Also introduces a new loss which encourages meaningful object-like representations.

Strengths: Paper is nice and clear, experiments give a satisfying demonstration of the effectiveness of each component of the architecture, experiments performed in a reasonably wide range of settings.

Weaknesses: Feels bit incremental. Relation networks/graph networks are everywhere these days, in a way this is just patching a relation network into BlockGAN, while at the same time stepping down from 3D to 2D, reducing complexity. It would be interesting (and more impactful) to see whether your improvements also help when transfered to BlockGAN (which is presumably better able to handle more complex scenes, scenes with occlusion, where reasoning in 3D matters, and also environments like that RealTraffic where the lack of 3D reasoning shows itself in other ways, as mentioned in the paper)

Correctness: As far as I can tell, yes.

Clarity: For the most part yes. There are a few grammatical issues here and there.

Relation to Prior Work: Yes.

Reproducibility: No

Additional Feedback: * Line 220: I didn't understand the explanation for why objects are rendered on the background for CLEVR and RealTraffic but not the other datasets. * Would be useful to show performance of a non-object-oriented GAN, as done in BlockGAN. * Would be interesting to come up with a way of qualitatively evaluating the object-decompositions in these object-oriented GANs, rather than just the reconstructions, but I realize that is a hard problem. * There is no mention of plans for releasing source code, please do release it.

[Author Response · NeurIPS 2020]

We thank the reviewers for their time and insightful feedback. Most appreciated the novelty of this work and the results. **R1.1: More realistic data.** See R3.6. **R1.2: Object appearance is not independent.** Our assumption is that the appearance *latent parameters* are independent (*i.e.*, choice of shape, color, texture and reflectance), not the *rendered appearance*. Interactions such as cast shadows and reflections *can* be captured by the generator network $G$. While these effects are subtle, some such interactions can be observed, *e.g.*, the shadows in row 2 objects 2&4 in Fig. 4r and row 1 image 3 in Fig. A7 (supp. mat.). **R1.3: Beyond small objects.** It is possible to handle much bigger objects. The only assumption is that objects are *smaller* than the scene, not *much* smaller. This parameter can be controlled by varying $H$, see also Fig. A7&8. **R1.4: Beyond centroid position.** This is beyond this work, but we are testing this extension by using other composition strategies. **R1.5: More varied interactions.** See R1.2: The generator can account for complex visual interactions. **R1.6: Iterative corrections.** This is possible, but empirically for our data a single feed-forward pass through the interaction module is sufficient. **R1.7: Predicting all at once?** As the primary purpose is regularization, we chose the simpler solution that avoids challenges, such as dealing with order, count, occlusion, *etc*. **R1.8: Didn't really change only one object.** Though very small, this effect is the result of composition at feature level. It could be mitigated by increasing the latent resolution $H$ or with a different composition strategy. **R1.9: Account for intersection.** The model pushes objects apart when appropriate according to the data statistics. In datasets such as CLEVR, the model does allow occlusions (see Fig. A6,7,8) because they occur in the data, while still preventing unrealistic cases such as objects at the same exact positions or implausible configurations. **R1.10: L.218 BlockGAN failure mode.** See R2.3. We also model interaction between *objects and background* (Eq. 1).

**R2.1/2/5: Training robustness; Disentanglement: better measures (MIG.)** RELATE *always* converged during training and, over 4 runs, $std(FID) < 5$ for *each* dataset of Tab. 2 except Shapestacks (7.9). A better metric than FID to measure disentanglement is possible, but computing the MIG score is not applicable in our case since our model does not feature an inference component. As an alternative, we toggle each object individually, looking at how the generated image changes. We then report the distance between the pixel location corresponding to the maximum image change and the location (scaled $\theta_i$) of the object that was toggled. The median distance is in favor of our model: 17/19/17/23/26 *vs.* BlockGAN's 19/18/272/22/98 for CLEVR5/CLEVR/ShapeStacks/Cars/Balls. Note that Balls has a bigger offset due to bigger latent object size (Tab. A1, $H'=8$). **R2.3/6 Ablation study inconclusive; BlockGAN fails BallsInBowl, why?** We ran ablations of Tab. 1 *three times* with same hyper-parameters (see A2/A3 in supp. mat.) and found that *only* our model was able to converge. Since BlockGAN does not account for background-object interactions (as we do in Eq. 1, L.134), it cannot place the balls within the bowl unless these are all predicted as part of the same (background) component. Similarly 'w/o pos. reg.' shows that $\Gamma$ is not sufficient: we need $P$ to force individual objects to appear. Finally 'w/o res.' shows that addition to $\hat{\theta}_k$ forces variance of object position and stabilizes training therefore enabling better disentanglement. **R2.4: Position regularizer vs shift.** It forces the object to appear; a shift alone is not sufficient because the network can still learn to not show individual objects (see R2.3). **R2.7: Position prediction accuracy?** Using the position predictor, RELATE reconstructs position with 11.3 mean pixel errors in Shapestack, which is quite good. **R2.8: Shapestack falling towers.** Proper modelling of falling towers requires modelling the 3D rotations of blocks and their angular dynamics, which the current version of the model does not handle (see conclusions). **R2.9: Physical interpretability?** We do model physically plausible configurations of objects and background. Extending the model to explicitly reason about dynamics is a next step.

**R3.1: Value of $K$** As noted in L.140, we do *not* need to specify the value of $K$, but rather a distribution of values in a reasonable range. In practice, the latter can be a rough guess (see R3.6). **R3.2: Perspective scenes.** CLEVR and RealTraffic are perspective scenes (see also Fig. 5 right: objects are changing sizes when being moved). As noted in L.220, perspective is accounted for when composing with the background. **R3.3/R4.5: Processing details/code.** Most of the requested details are given in the supp. mat. A4. The dataset/code will also be publicly released. **R3.4: L.100. Background vs foreground objects.** If an object (*e.g.*, tree) is fixed, the network can account for the resulting occlusions when composing objects with the background choosing not to render an object based on its position. **R3.5: BlockGAN vs RELATE FID score.** Thanks to our correlation module and position loss, RELATE does not generate unrealistic scenes, such as two objects intersecting or objects not on the ground. It also helps the generator at training time resulting in better FID scores by preventing mode collapse (see BlockGAN ablation). **R3.6: Experiment on GRAM [1]** The GRAM datasets have at most 24k images which is generally not enough to train a GAN and get a robust FID score.

Nonetheless, we trained from scratch on MH-30-HD (cropped to 512x512) with number of objects sampled in [1,8]. We obtained 117.8 FID on 400 test images and good object factoring (see figures), quality would undoubtedly improve with more data.

**R4.1: Incremental.** We introduce two innovations, position correlation $\Gamma$ and position regularization $P$, which enable training of our model on more complex data than the original BlockGAN. Empirically, these modifications are *necessary* for it to converge and/or to result in proper disentanglement. **R4.2: L.220.** This line does not state that objects are rendered *in* the background layer, but that they *use* information in this layer for rendering: *i.e.*, the final object appearance depends on its position w.r.t. the background due to the perspective effect. **R4.3: non-object oriented GAN** We trained {DRA,DC}-GAN on 64x64 images. Best FID scores: 80.8/84.4/108./57.2/38.8, with dataset ordering of Tab. 2. **R4.4: Object decomposition score.** We address this issue in R2.1/2/5.

[Meta-Review · NeurIPS 2020]

This paper studies the interesting but difficult problem of generating physically plausible scenes and videos which contain multiple objects that interact with each other. The generative model is formulated based on GAN by extending a 2D variant of BlockGAN with a relational layer. The proposed model includes a correction network to model object interactions and a structural loss function to evaluate the object poses. The reviewers and AC have read the author feedback carefully in addition to all the reviews during the discussion period. Although the original reviews somewhat diverged, the discussion has helped to converge their views. The proposed method is a combination of BlockGAN and ideas from graph neural networks. Considering the difficulty of the problem studied, the results are quite impressive in generating scenes and videos involving multiple interacting objects. Nevertheless, there are a number of concerns raised by the reviewers, including the issue of physical plausibility and the partial independence assumption. The paper also has room for improvement in its experiments. It is not clear whether the proposed method still works well for more realistic scenes involving more complex and variable interactions among the objects. Three of the four reviewers changed their overall scores during the discussion. Eventually all four reviewers are supportive of accepting this paper. Consequently, it is recommended by the AC to accept the paper for poster presentation. The reviewers have made a number of suggestions to improve the paper. The authors are recommended to consider them seriously for the revision.